# Liquid CO_2_-Capture Technologies: A Review

**DOI:** 10.3390/nano14231910

**Published:** 2024-11-28

**Authors:** Jie Zhu, Haokun Zhang, Tingting Li, Tingting Deng, Hao Zou, Yongqi Li, Dingyu Yang

**Affiliations:** 1College of Food and Biological Engineering, Chengdu University, Chengdu 610106, China; tthh1999@163.com (T.L.); dengtingtingcdu@163.com (T.D.); a3184985516@163.com (H.Z.); liyongqi20020223@163.com (Y.L.); 2School of Mechanical Engineering, Chengdu University, Chengdu 610106, China; zhkcs635128168@163.com; 3College of Optoelectronic Technology, Chengdu University of Information Technology, Chengdu 610225, China

**Keywords:** CO_2_ capture, liquid absorbent, alcohol-amine solution, nanofluid, gas–liquid mass transfer

## Abstract

Escalating global carbon dioxide (CO_2_) emissions have significantly exacerbated the climate impact, necessitating imperative advancements in CO_2_-capture technology. Liquid absorbents have received considerable attention in carbon capture for engineering applications, due to their high flexibility, reliability, and recyclability. Nonetheless, the existing technologies of liquid CO_2_ capture suffer from various issues that cannot be ignored, such as corrosion, elevated costs, and pronounced secondary pollution. More efforts are required to realize process optimization and novel absorbent innovation. This review presents nanofluids and other novel liquid absorbents such as ionic liquids, amino acids, and phase-change absorbents. The preparation, mechanisms of action, and influencing factors of nanofluid absorbents are discussed in detail to provide researchers with a comprehensive understanding of their potential applications. Further, the challenges (including energy loss, environmental and human health, barriers to application and capture performance, etc.) encountered by these innovative absorbents and techniques are also commented on. This facilitates side-by-side comparisons by researchers.

## 1. Introduction

With the advancement of industrial innovation and the intensification of global energy consumption, the rise in carbon dioxide emissions across various industries has heightened concerns about climate change and resource reserves. According to the “CO_2_ Emissions in 2022” report published by the International Energy Agency (IEA), global energy-related CO_2_ emissions surpassed 3.68 billion tons in 2022, marking a 0.9% increase compared to 2021. Furthermore, emissions from emerging markets and developing economies in Asia (excluding China) witnessed a notable increase of 4.2% [1,2]. In the face of this substantial growth, various carbon capture and storage (CCS) technologies have become available for decreasing CO_2_ emissions to mitigate global warming and secondary climate issues.

Since the 1930s, the utilization of monoethanolamine (MEA) has been documented for separating acidic gases [3] Contemporary research studies have expanded the number of liquid CO_2_-capture options, such as ionic liquids [4], phase-change solvents [5], and nanofluids [6]. However, the application of liquid CO_2_-capture technologies has given rise to associated challenges, including the suboptimal absorption efficiency of absorbents [7], heightened energy consumption [8] during the regeneration process, and a substantial environmental impact [9] linked to absorbent contamination. Extensive research efforts have been devoted to these challenges. Lu et al. [10] introduced polar solvents (DMF/DMSO/NMF) into the AMP (2-amino-2-methyl-1-propanol)-ethylene glycol system. This formulation enhanced the absorption capacity of the absorbent by 28.4% compared to the AMP aqueous solution. Smerigan et al. [11] developed a new highly efficient bio-based CO_2_ absorbent, which is composed of microalgal amino acid salts, thereby having an extremely low environmental impact.

With the rapid development of nanocatalytic materials and supramolecular chemistry, nanofluids have been invented and gradually become one of the potentially absorbents with liquid-phase mass-transfer enhancement. The main focus of this paper is to summarize the preparation of nanofluids, the enhancement mechanisms in gas–liquid mass transfer, and recovery methods, while also discussing the development trends of nanofluids in the field of carbon dioxide capture. Additionally, other absorbents such as ionic liquids, phase-change solvents, and amino acid salt solutions are briefly introduced.

## 2. Conventional Amine Capture Technologies

The MEA solution is one of the most traditional and mature CO_2_ absorbents. In the post-combustion capture of the power plant, the MEA-based CO_2_ technological process is shown in Figure 1. MEA embraces many advantages as an absorbent, mainly including the high theoretical reaction ratio and its multifunctionality for trapping a variety of pollutants [12].

As far as we know, the overall reaction of CO_2_ with aqueous MEA can be expressed as follows [13]:(1)$$CO_{2}+2RNH_{2}=RNHCOO^{-}+RNH_{3}^{+}$$

Caplow’s seminal work [14] verifies a theory that the most probable early event in the capture from aqueous MEA is the formation of a zwitterion. This theory is in accordance with the analyses of the kinetics of the reactions [15,16,17]. Moreover, the carbamate is the prevailing product of the CO_2_ interaction with the MEA solution, the decomposition of which requires considerable energy. In summary, the two stages of reactions implied in the zwitterion mechanism can be written as follows [18]:(2)$$CO_{2}+RNH_{2}=RNH_{2}^{+}COO^{-}$$
(3)$$RNH_{2}^{+}COO^{-}+B=RNHCOO^{-}+BH^{+}$$
where B represents the alkali component.

### 2.1. Mixed Amine Solution Absorption Technique

The absorption method using mixed amine solutions has been investigated as a superior alternative to single-component amine solutions in the CO_2_-capture process, aimed at overcoming their limitations and enhancing capture efficiency. Commonly used improvement methods usually refer to mixing a primary or secondary amine with a tertiary or sterically hindered amine [19] to give the absorbent the properties of the above components. Thus, the mixed solution possesses both a high CO_2_-absorption capacity and an excellent regeneration effect.

For example, He et al. [20] compared four amine-based blends and found that the blend of AMP (2-amino-2-methyl-1-propanol) with 2DMA2M1P (N,N-dimethylacetamide-2-methyl-1-propanamine) exhibited a higher desorption rate and lower energy consumption. BZA (benzylamine) + 2DMA2M1P showed an excellent absorption rate, attributed to AMP and BZA having the highest equilibrium solubility and second-order reaction rate among them. These studies demonstrated that mixed amines represented by AMP have an unstable steric hindrance, which inhibits the formation of carbamate and thus promotes the reaction [21]. Ge et al. [7] investigated seven different polyamines containing varying numbers of secondary amine and primary amine groups. They directly measured the heat of CO_2_ absorption using a high-precision microcalorimeter and compared their cyclic capacities to evaluate the CO_2_ cyclic capture times, absorption rate, and desorption rate. The experiment result showed that (1) the higher proportion of primary amino functional groups in the molecule, the higher the CO_2_ absorption heat; and (2) an increase in the proportion of secondary amino functional groups in the molecule causes a quicker CO_2_ desorption rate and encourages more complete desorption progress. This conclusion on exploring the factors influencing the heat of absorption can help to reduce the cost of mixed amine applications.

### 2.2. Sterically Hindered Amine

The amine absorption capacity is influenced by the spatial structure of the amine molecule. The initial reaction between the sterically hindered amine and CO_2_ still follows the zwitterionic theory; however, the steric hindrance renders the resulting carbamate unstable, leading to its rapid hydrolysis into bicarbonate and free amine. Furthermore, the free amine produced from this hydrolysis can continuously react with carbon dioxide, facilitating ongoing CO_2_ consumption [22,23]. This dynamic process highlights that the interplay between steric effects and reaction stability is important in the context of carbon capture or related applications. Sterically hindered amines [24] and cyclic amines [25] are widely studied in experimental environments. In Kim et al. [26], the effects of the molecular structure of amine on CO_2_ loading, cyclic capacity, the absorption–desorption rate, and pKa were investigated, and promising candidates for CO_2_ capture, along with an amine blending strategy, were proposed. The general formula for the reaction is as follows:(4)$$\mathrm{AmineH}\;+\;H_{2}O\;+\;CO_{2}\;=\;\mathrm{Amine}H_{2}^{+}\;+\;\mathrm{HC}O_{3}^{-}$$

## 3. New Liquid CO_2_ Absorbents and Techniques

Amine absorbents have a high absorption capacity, while some disadvantages limit their development, such as high energy consumption during regeneration, corrosiveness, and degradation. Researchers have carried out various optimizations for amine-based solvents in the CO_2_-capture process, including fractal reactors [27] and the use of liquid membranes [28] for absorption. However, a study reported that the CO_2_-capture cost for a newly constructed power plant is CNY 150 per tonne, whereas, for an existing CCS retrofitted power plant, this cost ranges from CNY 162 to 185 per tonne [29]. Large-scale industrial equipment optimization is clearly not cost-effective. In order to promote industrial applications, it is necessary to develop new absorbers with better performance than amine-based absorbents, which include amine-based solutions, ionic liquid, amino acid salt, and phase-change absorbents. In comparison with the previous items, nanofluid is a better choice. Nanofluids offer a promising solution by enhancing the liquid-phase mass-transfer process without requiring modifications to existing facilities. Adding nanoparticles to solvents could improve CO_2_ absorption performance and reduce energy requirements.

### 3.1. Nanofluid

Choi first introduced the concept of nanofluids at the Argonne National Laboratory in the United States [30]. The nanofluid was initially applied in the field of heat transfer and applied to gas-separation applications after the early 21st century. Adding solid nanoparticles of the third dispersed phase into the absorbents is an important method used to enhance gas-separation efficiency. This way is also helpful in enhancing capture performance and reducing the vapor pressure of the absorbent. According to the research by Irani et al. [31], it was shown that a 9.1% increase in CO_2_ solubility was achieved via adding 0.1% graphene oxide (GO) into a 40% solution of N-methyl diethanolamine (MDEA), with no significantly higher absorption capacity when 0.2 wt. % GO was added to solutions than at the 0.1% addition level. The solubility of CO_2_ increases because the oxygen-containing groups on GO provide a wide range of reaction sites and interlayer space for gas adsorption. Irani stated that this enhancement has a reverse and direct relationship with increasing temperature and pressure. Some research has demonstrated different improvements in mass-transfer effects with the use of different types of the third dispersed phase [32]. The enhancing effect of TiO_2_ is more prominent in SiO_2_, Al_2_O_3_, and TiO_2_ [33]. Under certain conditions, the surface properties [34] and photocatalytic characteristics [35] of TiO_2_ may enhance the mass-transfer pathways at the gas–liquid interface. Another important factor is the size of the nanoparticles. Darvanjooghi et al. [36] investigated the effects of nanoparticle size on carbon dioxide (CO_2_) absorption in silica/water nanofluid by use of a bubble column absorption system, and the results showed that the surface renewal rate increased with the decreasing of nanoparticle size. However, nanoparticles of too small a size may have other negative effects, such as dispersion defects.

Some difficulties also hinder the development of nanofluid applications. The poor dispersion stability of nanofluids has been considered a long-existing issue that limits their further development and practical application. A prevalent strategy to mitigate this issue involves the introduction of dispersants and surfactants. The use of surfactants can improve the charge or functional groups on the surface of nanoparticles. However, it is imperative to recognize that the presence of dispersants and surfactants may influence the mass-transfer coefficient and surface tension of the liquid phase in gas–liquid mass-transfer processes [37,38]. Furthermore, these dispersants and surfactants may pose a threat to the environment [39] and increase the cost of manufacturing nanofluids.

On the other hand, the recovery of nanoparticles from nanofluids is an inevitable practice to reduce production costs. Nanoparticles have intrinsic properties, such as particle size, density, magnetic properties, electric properties, and aggregation tendency. These properties could be used to fractionate nanoparticles using external fields, such as centrifugal force or an electric field. These two external forces have been commonly applied for nanoparticle sorting, namely ultracentrifugation and gel electrophoresis [40]. The following text will provide a detailed introduction to the preparation methods, enhanced gas–liquid mass-transfer mechanisms, stability, recovery methods, and modification of nanofluids.

#### 3.1.1. Synthesis of Nanofluids

The preparation process of nanofluids exerts a significant influence on the agglomeration of particles and other pertinent properties [41]. To our knowledge, nanofluids are produced by the uniform dispersion of solid nanoparticles into different base fluids. The synthesis of nanofluids can be divided into two methods, namely the single-step method and the two-step method, as presented in Figure 2.

In the two-step technique, the steps of preparing granules and dispersing granules are split up. Many researchers have prepared nanofluids with this method due to its simplicity and suitability [42]. However, aggregation and clustering will inevitably occur due to the high surface energy of the nanoparticles. The issue of particle dispersion and reduction in agglomeration in nanoparticles can be effectively addressed through the application of physical or chemical methods. For instance, a three-component ZnO+Al_2_O_3_+TiO_2_/water-based composite nanofluid was prepared by Ahmed et al. [43] using a two-step process and ultrasonic dispersion. The use of ultrasound to uniformly disperse nanoparticles into the liquid phase is a simple and reliable physical method commonly used in laboratories. Some researchers have synthesized silver nanoparticles of different sizes by changing temperature conditions in the preparation process [44] and summarized the law that particle size increases with increasing temperature. This law helps the research process of synthesizing nanoparticles of specified sizes.

In the one-step method, nanoparticles are produced and suspended in a base fluid directly to avoid drying and aid in the storage, transportation, and dispersion of nanoparticles. Thus, the purity of the nanoparticles and the stability of fluids are guaranteed. Aberoumand et al. [45] prepared nanofluid by a one-step method of the electrical explosion of wire (EEW), and this nanofluid could remain stable for several months. The method generates nanoscale particles or droplets by spraying or atomizing a liquid under a high electric field strength. It has good generalizability and environmental friendliness. Under normal circumstances, the purity and stability of nanofluids prepared by the one-step method are better than those prepared by the two-step method.

#### 3.1.2. Mechanisms

An exact comprehensive mechanism for the enhancement of mass transfer by nanoparticles has not yet been established. Studies have proposed three major models, which are widely accepted by the scientific community.

(1) Shuttle mechanism: The schematic diagram is presented in Figure 3. Nanoparticles adsorb a certain amount of mass-transfer components. Subsequently, the gas adsorbed by the nanoparticles enters the liquid phase. Under the influence of the concentration gradient around the components, significant desorption phenomena occur in the vicinity of the nanoparticles. This results in nanoparticle repositioning and re-exposure to the vapor-phase components. The effect of the shuttle phenomenon can be characterized by measuring the diffusion coefficient (*D*co_2_) within the liquid phase [46].

(2) Fluid dynamics mechanism: Brownian motion and micro-convection are the foundation of the hydrodynamic mechanism. According to the two-film theory, the primary resistances in gas–liquid mass transfer encompass gas film resistance, interfacial resistance, and liquid film resistance. These three kinds of resistance are all impacted by Brownian motion in the gas–liquid interface, micro convections, and velocity disturbances caused by nanoparticles. The schematic diagram is shown in Figure 4, the liquid-phase mass-transfer resistance can be weakened due to the reduction in the thickness of the boundary layer. The hydrodynamic effect in CO_2_ absorption was demonstrated by Lee et al. [47] using the shadowgraph method. In the visualization results at 8 s, the plume reconstituted more actively in the nanofluids as compared with pure methanol. The diffusion coefficient was calculated from the visualized test results, and the result indicates that fluid dynamic effects can enhance the mass transfer of nanofluids. However, some researchers disagree with the mechanism. The impact of SiO_2_ nanoparticles on the mass transfer of O_2_ and NaCl was investigated by Feng et al. [48]. It was observed that in the presence of nanoparticles, mass-transfer enhancement did not occur. The authors concluded that the promotion of mass transfer by nanoparticles through Brownian motion and microscale convective effects was not evident.

(3) Mechanisms of bubble aggregation inhibition: This mechanism is commonly used to explain the phenomenon of compositional exchange of bubbles in liquids. The schematic diagram is shown in Figure 5. The mass transfer between gas and liquid is highly pronounced around bubbles in the liquid phase, while the bubbles in nanofluids have smaller volumes and shorter lifetimes than those in ordinary liquids. The main reason for this type of phenomenon is that nanoparticles attach to the surface of large bubbles and collide with each other, causing the bubbles to burst, and this behavior of nanoparticles is called the ultra-small size effect. There are five main physical properties affecting the behavior of nanoparticles, including the surface tension of the solution, the density difference between particle and solution, the solution viscosity, the solid particle size, and the hydrophobicity of particles. More importantly, the Laplace–Young equation explains the cause of bubble rupture in terms of size; bubble pressure is directly proportional to bubble size. The Kelvin equation explains the relationship between the internal pressure and solubility of the bubble and relates it to the force of mass transfer [49]. This equation explains the phenomenon that the solubility of the gas increases with an increase in the bubble’s internal pressure, thus leading to an increase in the driving force for mass transfer. 

In conclusion, the aforementioned three theories regarding the enhancement of mass transfer in nanofluids can reasonably explain the phenomenon of enhanced mass transfer in nanofluids within a certain range. It is speculated by some researchers that the actual mechanism may be a combination of two or more mainstream theories, which requires in-depth study.

#### 3.1.3. Dispersion Stability of Nanofluids

The dispersed state of nanofluids depends on the microscopic forces that are exerted on the nanoparticles, such as van der Waals forces and electrostatic forces [50]. The magnitude of forces between particles depends on the distance; the tendency of nanoparticles to aggregate and precipitate often occurs in short distances and is also affected by gravity. 

In addition to the interparticle distance, the shape of the nanoparticles may also lead to aggregation and precipitation phenomena. By comparison, the neighboring rod-shaped nanoparticles have a larger contact area than the spherical nanoparticles. Thus, rod-shaped nanoparticles have a stronger attraction between neighbors and a stronger tendency to form aggregates [51].

Some factors also contribute to the reduced likelihood of aggregation, such as shear flow and irradiation [52]. Finally, the settling force of the nanofluid is calculated by Equation (5), and the viscous resistance during motion by Equation (6) [53]:(5)$$F_{d}=\frac{\pi d^{3}(\rho_{p}-\rho_{l})g}{6}$$
(6)$$F_{r}=3\pi \mu d\mu_{0}$$
where *F*_d_ represents the settling force, and *F*_r_ is the viscous resistance. *μ*_0_ is the settling velocity, *μ* is the dynamic viscosity, *ρ*_p_ is the density of the nanoparticles, and *ρ*_l_ is the density of the base fluid.

The stability of nanofluids is related to the velocity of Brownian motion; some big nanoparticle clusters break into smaller ones and reduce the flow resistance of particles in nanofluids [54]. In the high-temperature regeneration of absorbents, this low resistance leads to a significant increase in heat-transfer efficiency. The temperature gradient in the vertical direction of the regeneration tower is particularly pronounced, thereby enhancing the micro-convective movement of the fluid, which is beneficial for the regeneration of the absorbents [55,56]. 

#### 3.1.4. Modification

Surface modification refers to functionalizing the nanoparticle surface, which is known as a promising technique for expanding the different functions of nanofluids (as shown in Figure 6). Nanofluids with functionalized nanoparticles offer excellent physical and chemical properties with low pollution [57]. For instance, the NH_2_-rGO (amine-functionalized reduced graphene oxide)/MDEA nanofluid synthesized by Vahid et al. [58] has a richer reaction potential and results in a 16.2% enhancement of mass transfer compared to the rGO/MDEA nanofluid. To further minimize the environmental impact, researchers frequently choose environmentally friendly functional groups, such as amino acids, which are then grafted onto nanoparticles. The synthesis and preparation of Fe_3_O_4_-proline, Fe_3_O_4_-lysine, and Fe_3_O_4_@SiO_2_-NH_2_ nanofluids were conducted by Elhambakhsh et al. [59] to enhance CO_2_ absorption. Fe_3_O_4_-proline, Fe_3_O_4_-lysine, and Fe_3_O_4_@SiO_2_-NH_2_ nanoparticles showed a 9.6%, 14.83%, and 17.61% higher uptake capacity, respectively, compared to Fe_3_O_4_ nanofluids. However, it is noteworthy that the number of active sites in the nanofluid tends to decrease with an increasing number of cycles, leading to a reduction in the CO_2_-absorption performance of the modified nanofluid after multiple cycles.

There are another set of factors influencing the enhancement effects of nanofluids, such as the space structure of the group, the type of nanoparticles, the concentration of nanofluids [60], and the type of base fluid [61]. Symmetric branched amino functional groups were synthesized by Arshadi et al. [62] and incorporated onto Fe_3_O_4_@SiO_2_, yielding Fe_3_O_4_@SiO_2_-NH_2_ nanoparticles. The CO_2_ absorption capacity of Fe_3_O_4_@SiO_2_-NH_2_ was observed to increase by 37.3% compared to Fe_3_O_4_, with a slightly higher CO_2_-capture rate. This enhancement can be attributed to the higher nucleophilic reactivity and density of active sites for Fe_3_O_4_@SiO_2_-NH_2_, facilitating better chemical bonding with CO_2_. It can be speculated that optimizing the structure of the functional groups and increasing active sites could potentially be the direction or focal point of future research on modified nanoparticles.

In addition, the stability of nanofluids can be enhanced through modification. Zhang et al. [63] used a silane coupling agent (APTS) to modify nanoscale TiO_2_. The principle of this behavior lies in the binding of APTS with hydroxyl groups on the surface of TiO_2_ nanoparticles, resulting in a spatial hindrance effect that prevents the agglomeration of the nanoparticles. Han et al. [64] prepared surface-modified gold nanoparticles and dispersed them into water with polyethylene glycol (PEG) or polyvinylpyrrolidone (PVP) polymers, and succeeded in improving the stability of gold nanoparticles. PEG and PVP as hydrophilic polymers may enhance the hydrophilicity of gold nanoparticles to a certain degree.

#### 3.1.5. Recycling

The recycling and reuse of nanofluids can mitigate environmental pollution and reduce costs. Dialysis and ultrafiltration are widely utilized in laboratory settings because of their crucial role in testing small-scale recycled nanoparticles. However, the shortcomings of the above methods are a long processing duration and high costs, which would be exposed on a larger industrial scale [65]. Centrifugation is another reliable separation method. However, nanoparticles were confirmed to have the tendency to aggregate randomly under gravity conditions (centrifugal condition) by the study of Tsuchiya [66]. For example, gold nanoparticles tend to be assembled into linear clusters under high gravitational acceleration in centrifugal forces [67]. The presence of linear clusters shortens the stable existence of nanofluids, which is not conducive to preservation and transportation.

Magnetic recovery is considered to be a potential recovery route compared to centrifugation. By exploiting the magnetic properties of Fe_3_O_4_, Fe_3_O_4_@PDA nanoparticles have been successfully recovered via a magnetic field [68]; this provides an effective and efficient method for converting nanoparticle dispersions into a stable dry powder form. The separation of magnetic particles is facilitated by magnetic separation techniques that manipulate magnetic flux and field gradients. There are primarily three technical approaches for the isolation of nanoparticles from liquid phases: High-Gradient Magnetic Separation (HGMS), Magnetic Field Flow Fractionation (MFFF), and Electric Field Flow Fractionation (EFFF). HGMS is an analytical technique used to isolate magnetic species from a nonmagnetic environment utilizing column flow. It was originally proposed to serve mineral beneficiation, water and waste treatment, and chemical processing, as well as the separation of micron-size magnetic particles. HGMS provides a way to capture magnetic material in column flow by dehomogenizing the magnetic field through the column to create a large field gradient. MFFF has been reported as one of the most promising approaches to separation based on the application of external magnetic fields to initiate nanoparticle magnetization. EFFF is another innovative technology that can be utilized primarily for the separation of water/nanoparticle dispersions in narrow channels through the application of an electric field. In this method, the magnetic nanoparticles are separated based on their electrophoretic mobilities and sizes [69]. The magnetic recovery approach is only applicable to magnetic nanofluids, and for the recovery of nonmagnetic nanofluids, the most promising technology remains centrifugal.

#### 3.1.6. Negative Environmental and Health Impacts

The hazards of nanomaterials including nanofluids and nanoparticles to the human body and the environment cannot be ignored. The main health effects of nanomaterials are inflammation, allergy, genotoxicity, and carcinogenicity. In addition, nanomaterials have toxic effects on development and reproduction, including fetal development, the central nervous system, the reproductive system, and the immune system [70]. 

Nanoparticles released into the environment combine with natural colloidal substances to form mixtures, which are deposited into soil and water and eventually absorbed or ingested by humans. The use of non-toxic nanoparticles has been proposed by researchers as a way to mitigate the risks of nanoparticles to humans and the environment. An approach to adapting the existing regulatory framework has been adopted by the EU to address the nano-form issue [71]. Along with further research on nanofluids, it is believed that more comprehensive regulations will be proposed to limit the misuse of nanofluids.

### 3.2. Other New Liquid Absorbents and Techniques

#### 3.2.1. Ionic Liquids

Ionic liquids, composed entirely of ions and existing in liquid form at room temperature, exhibit strong electrostatic interactions and hydrogen bonding, which enhance the solubility of CO_2_ [72]. Ionic liquids are characterized by a low vapor pressure, low reaction enthalpy [73], wide liquid range, and high thermal stability. These advantages of ionic liquids have great potential for application in the field of carbon capture. However, the absorption capacity of conventional ionic liquids is lower than that of alcohol amines. The current solution given by researchers is to add an amine moiety to the ionic liquid. Furthermore, the adjustability of ionic liquids suggests that the chances for preparing a broad array of ionic liquids with ions incorporating functional groups are rather good and functionalized ionic liquids have been studied since 2002 [74]. According to the different types of ions at the CO_2_-philic sites, functional ionic liquids can be divided into cation-functionalized ionic liquids, anion-functionalized ionic liquids, and cation–anion dual-functionalized ionic liquids [75]. Compared with conventional ionic liquids absorbing CO_2_, functionalized ionic liquids or task-specific ionic liquids could chemically absorb CO_2_ through single-site mechanisms or multiple-site mechanisms. Unfortunately, the application of these functionalized ionic liquids in CO_2_ capture is compromised by their high viscosity [76] and high cost [77].

#### 3.2.2. Amino Acids

The absorption mechanism of amino acid absorbents is analogous to that of primary and secondary amines [78]. In comparison with the intensively investigated MEA, amino acids exhibit distinctive advantages as compared with the intensively investigated MEA. Amino acid salts are noteworthy for their low energy consumption, as well as their safety and environmental friendliness [79]. The CO_2_ absorption heat of K-Lys (lysine potassium salt) was estimated by Zhao [80] and others using the Gibbs–Helmholtz equation. In comparison with a 30 wt% MEA solution (84.5 kJ/mol), lower energy is represented by 20–30 wt% K-Lys (55–70 kJ/mol). Additionally, Rouzbeh et al. [81] summarized the toxicity of amine compounds revealing an order of PZ > MEA > MDEA, with all amino acids exhibiting lower toxicity. The non-toxic and non-hazardous nature of amino acids allows them to be useful in certain specific locations (e.g., hospitals and kindergartens) and areas. However, Erga et al. [82] pointed out that glycine produces unpleasant odors during the regeneration process, so it is necessary to install purification facilities when using some amino acid absorbents. This will increase the cost of CO_2_ capture and hinder the large-scale application of amino acid absorbents. 

#### 3.2.3. Phase-Change Solvents

The concept of phase-change absorbents was first proposed by Liang Hu [83] at Hampton University. Phase-change solvents are homogeneous (single-phase) solvents under normal conditions, but undergo a phase transition into a heterogenic (two-phase) system, triggered by changes in polarity, hydrophilicity, ionic strength, or hydrogen bond strength to form a CO_2_-lean liquid phase and a CO_2_-enriched liquid or solid phase. In contrast with traditional MEA absorbents, phase-change absorbents possess characteristics such as a low regeneration temperature and high reaction rates [84]. Kim and Lee [85] revealed that the CO_2_ loading in the rich phase was a critical factor influencing the energy demand for the generic biphasic solvent-based process. The energy penalty decreased with increasing CO_2_ loading in the rich phase. This feature is ideal for large-scale CO_2_-capture applications. A specific disadvantage of phase-change solvents is unavoidable, the liquid–liquid phase-separation requires. Additionally, some of the absorbent is still corrosive and has a low fault tolerance in each step due to the risk of incomplete phase separation which will affect the subsequent regeneration process [5]. 

## 4. Conclusions and Future Perspectives

The liquid CO_2_-capture technologies are considered to be one of the most useful CO_2_-capture routes because of their significant absorption capacity, high reliability, and rich engineering foundation. The main discussion throughout the text is summarized in points as follows:

(1) Nanofluids are one of the most valuable applications of liquid carbon dioxide-capture technology and show the ability to significantly improve mass transfer. The enhancement mechanism of nanofluids may be related to the shuttle effect, the hydrodynamic effect, and bubble aggregation inhibition. However, there are differences in the views of different researchers. There are also different conclusions about the enhancement effect of modified nanofluids. In addition, the effect of the nature of nanofluids on gas–liquid mass transfer has been widely discussed. At the application level, existing preparation and recovery methods do not fully meet industrial requirements, and innovative approaches need to be sought. In conclusion, nanofluids technology is a controversial and innovative liquid CO_2_-capture technology.

(2) Ionic liquids have excellent physical and chemical properties. Not only do they increase the solubility of CO_2_ significantly, but they also have a low energy requirement for regeneration. The anions and cations in ionic liquids offer a wealth of functionalization possibilities. However, the high viscosity and high cost of functionalized ionic liquids need to be considered.

(3) Amino acids and amine solutions have several points of identity. For example, both absorb CO_2_ by binding RNH_2_ groups, and the absorption is affected by the spatial structure and species. However, amino acids have superior application value due to their low toxicity and low pollution properties. With further development, they may be able to replace amine solutions.

(4) The phase-transition behavior of phase-transition absorption has two sides. On the one hand, the phase-change behavior effectively enriches CO_2_, resulting in an increase in the regeneration efficiency of the phase-change solvents. On the other hand, the phase-change behavior increases the complexity of the process. When designing the process, it is necessary to balance the energy, process complexity, equipment life, and efficiency according to the CO_2_ load.

A few outlooks for liquid CO_2_-capture technologies are as follows:

(1) The traditional amine solution CO_2_-capture technology has a certain engineering hardware basis, but its widespread adoption is hindered by absorbent capacity and rate and regeneration energy consumption. Researchers have developed mixed amine absorbents based on the characteristics of primary, secondary, and tertiary amines, which to some extent compensate for the drawbacks in absorption rate and the capacity of single amine absorbents. However, the use of alcohol amine solutions still faces drawbacks such as high energy consumption, strong corrosiveness, and significant pollution. Future work will focus on finding better combinations of amine systems with regard to the greater CO_2_ absorption activity, faster CO_2_ desorption rate, and lower regeneration heat duty. Also, amino acid salts, ammonia water, and other absorbents are new raw materials for absorbent combinations.

(2) Nanofluids offer the potential to improve mass-transfer efficiency and reduce energy consumption based on traditional amine-based CO_2_-capture technology. The practical usage of nanofluids in engineering applications would raise even harsher requirements on the function, stability, and environmental protection properties. In the future, the positive effects of different nanoparticle and functional groups on liquid carbon-capture technologies need to be considered. Secondly, based on their bonding principles, the bond strength could be increased to extend the circulation cycles of nanofluids. 

(3) Other novel liquid absorbents (including ionic liquids and phase-change absorbents) are of undeniable value. The application of ionic liquids is hampered by high viscosity and high cost. Reducing the cost and viscosity of ionic liquids while maintaining their excellent physicochemical properties is an area to focus on in the future. Secondly, the environmental and health impacts of ionic liquids need to be fully assessed in subsequent studies. Phase-change solvents have high requirements for processes and equipment. In practical applications, efficient and cost-effective solutions can be designed based on the specific conditions of the plant.

(4) CO_2_ absorption in solution is a complicated process involving gas–liquid or gas–liquid-solid mass transfer and chemical reactions. Exploring new absorption materials and revealing the kinetic limiting step is key to facilitating the carbon-capture process. Until now, novel liquid absorbents and techniques exhibit distinct advantages. For instance, amino acid salts demonstrate low pollution potential, ionic liquids exhibit higher physical solubility, and phase-change absorbents result in lower energy consumption. However, the kinetic law of the mass-transfer reaction has not been studied deeply, and this has greatly limited the improvement of carbon-capture technology because gas–liquid mass transfer is a phenomenon of multiple factors, and a single increase in the equilibrium constant of a chemical reaction can only enhance carbon-capture technology to a certain extent. Therefore, multi-factor theoretical research and better model design are essential.

## Figures and Tables

**Figure 1 nanomaterials-14-01910-f001:**
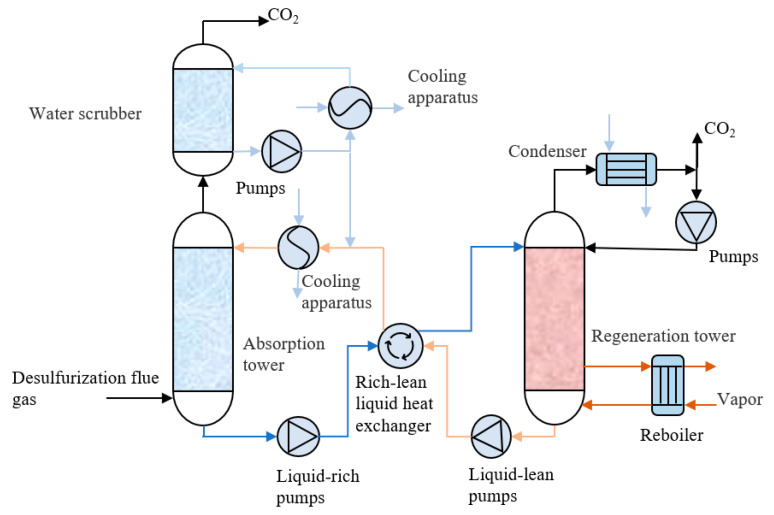
Process flow diagram of an amine absorption process.

**Figure 2 nanomaterials-14-01910-f002:**
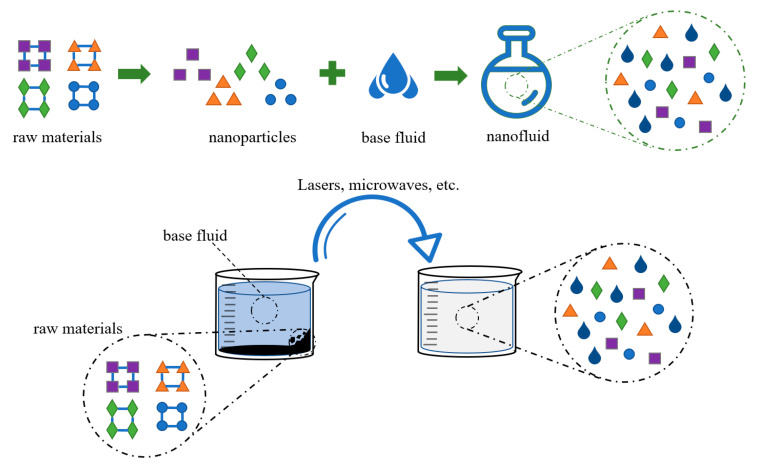
Single-step technique and two-step technique.

**Figure 3 nanomaterials-14-01910-f003:**
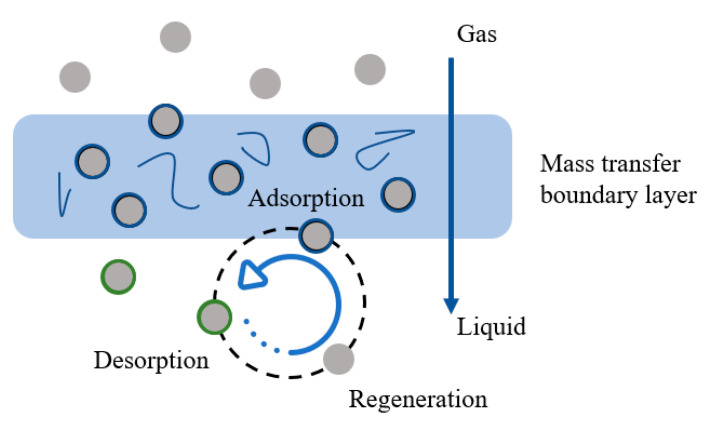
Schematic diagram of the shuttle mechanism.

**Figure 4 nanomaterials-14-01910-f004:**
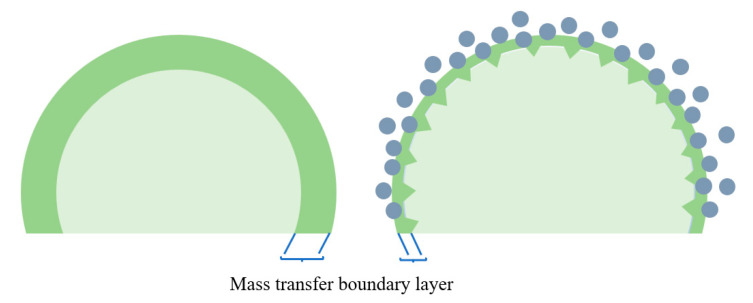
Schematic diagram of the hydrodynamic effect.

**Figure 5 nanomaterials-14-01910-f005:**
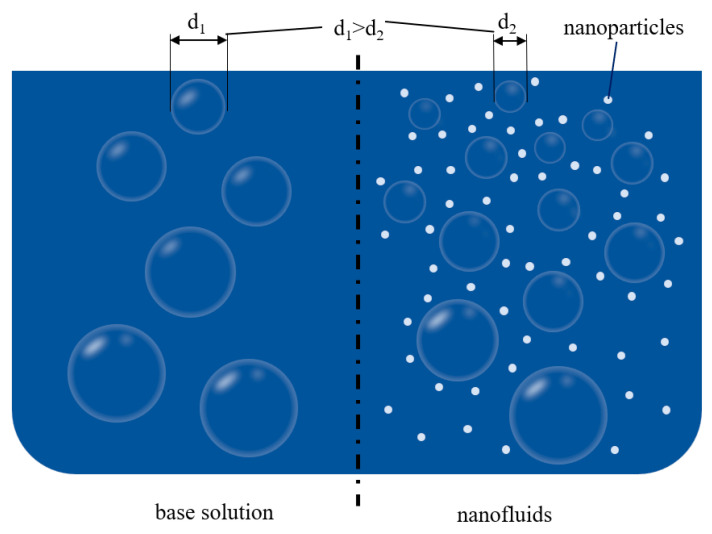
Schematic diagram of the mechanisms of bubble aggregation inhibition.

**Figure 6 nanomaterials-14-01910-f006:**
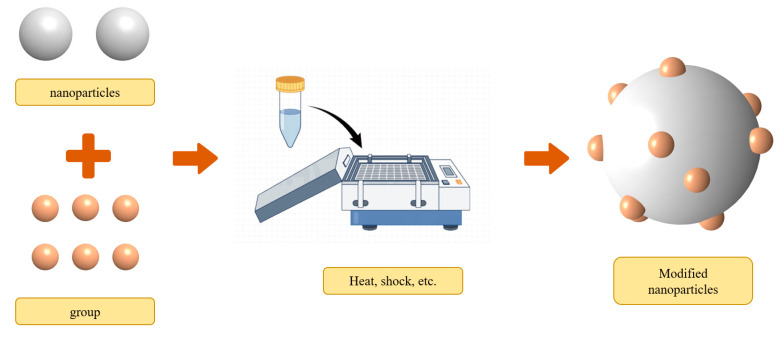
Typical modification method of nanoparticles.

## Data Availability

This manuscript does not report data generation or analysis.

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
