# Peer review of "Liquid CO2-Capture Technologies: A Review"

_nanomaterials, 2024, doi:10.3390/nano14231910_

Round 1
Reviewer 1 Report
Comments and Suggestions for Authors
Title: The liquid CO2 capture technologies: A Review
The Review must be reorganized and some other paragraph has to be included in the manuscript such as further investigations, the point of view of the authors and separately the summary of all informations reported in the review.
Below I reported just some example of not clear exaplanation of some senteces that has to be improved.
Line 28….widespread attention; What type pf attention? Rewrite better this sentece.
Line 36-37…such as ionic liquids, phase-change solvents, and nanofluids. Miss the reference.
Line 40 miss the reference
Line 42. Lu et al.[4] introduced polar solvents (DMF/DMSO/NMF) into the AMP (2-amino-methyl-1-propanol) - EG (ethylene glycol) system, resulting in an effective enhancement of CO2 absorptive capacity and a reduction in regeneration energy consumption.
Can you expliain this enhancement to respect of what and how muchi t was enhance?
Comments on the Quality of English LanguageThe english of the Review must be improved.
Reviewer 2 Report
Comments and Suggestions for Authors
The review entitled “The liquid CO2 capture technologies: A Review” describes the technologies of CO2 capture. I think that it can be considered for publication in Nanomaterials, but it requires major revision first.
1. Please, emphasize the goal of the review
2. As far as we know, the overall reaction of CO2 with aqueous MEA can be expressed as … Please, provide reference.
3. Line 130. Please, explain the role of graphene oxide in increasing CO2 solubility.
4. Is it possible to describe the mechanism of CO2 capture by ionic liquids?
5. Please, add the conclusion section
Round 2
Reviewer 1 Report
Comments and Suggestions for Authors
This article can be published in its current form.
Reviewer 2 Report
Comments and Suggestions for Authors
I think that the Review can be accepted in present form